# Prediction of Diabetes Mellitus Progression Using Supervised Machine Learning

**DOI:** 10.3390/s23104658

**Published:** 2023-05-11

**Authors:** Apoorva S. Chauhan, Mathew S. Varre, Kenneth Izuora, Mohamed B. Trabia, Janet S. Dufek

**Affiliations:** 1Department of Mechanical Engineering, University of Nevada, Las Vegas, NV 89154, USA; mohamed.trabia@unlv.edu; 2Department of Mechanical Engineering, University of Washington, Seattle, WA 98195, USA; msvarre@uw.edu; 3Department of Internal Medicine, University of Nevada, Las Vegas, NV 89154, USA; kenneth.izuora@unlv.edu; 4Department of Kinesiology and Nutrition Sciences, University of Nevada, Las Vegas, NV 89154, USA; janet.dufek@unlv.edu

**Keywords:** classification, prediction, dynamic plantar pressure, diabetic peripheral neuropathy, foot ulceration

## Abstract

Diabetic peripheral neuropathy (DN) is a serious complication of diabetes mellitus (DM) that can lead to foot ulceration and eventual amputation if not treated properly. Therefore, detecting DN early is important. This study presents an approach for diagnosing various stages of the progression of DM in lower extremities using machine learning to classify individuals with prediabetes (PD; n = 19), diabetes without (D; n = 62), and diabetes with peripheral neuropathy (DN; n = 29) based on dynamic pressure distribution collected using pressure-measuring insoles. Dynamic plantar pressure measurements were recorded bilaterally (60 Hz) for several steps during the support phase of walking while participants walked at self-selected speeds over a straight path. Pressure data were grouped and divided into three plantar regions: rearfoot, midfoot, and forefoot. For each region, peak plantar pressure, peak pressure gradient, and pressure–time integral were calculated. A variety of supervised machine learning algorithms were used to assess the performance of models trained using different combinations of pressure and non-pressure features to predict diagnoses. The effects of choosing various subsets of these features on the model’s accuracy were also considered. The best performing models produced accuracies between 94–100%, showing the proposed approach can be used to augment current diagnostic methods.

## 1. Introduction

Diabetes mellitus (DM) is a condition in which an individual is unable to either produce insulin, use naturally produced insulin correctly, or produce enough insulin to break down blood glucose [1]. The resulting high blood sugar levels can cause long-term organ damage. The World Health Organization estimates that diabetes is the seventh leading cause of death globally [2]. According to the American Diabetes Association, 11.3% of the U.S. population, or 37.3 million people, had DM in 2019 [3].

Health complications associated with diabetes include vision loss, heart disease, stroke, kidney failure, and amputation of lower extremities [4]. Those affected may also experience a higher mortality rate due to these health challenges [2]. Some of these complications can be related to peripheral neuropathy, which is the degradation of nerves in the extremities, and peripheral vascular disease, which causes reduced blood flow to the limbs [5]. The combined internal blood vessel damage coupled with the loss of sensation results in an increase in mechanical loads on the affected plantar tissue, leading to the development of plantar ulceration. These ulcerations, which are difficult to diagnose early, can lead to amputations if left untreated [5]. It was shown that DM was associated with more than half of the lower-limb amputations in the United States in 1991 [6]. It was also estimated that approximately 70% of all amputations were for people with DM [7] and about 60% of below-knee amputations were associated with DM [8]. While great strides were achieved in helping amputees cope with their losses, addressing the causes of amputation is highly desirable since approximately two million people in the United States live with limb loss. This number is expected to double by 2050 [9].

It has been reported that only 2% of the patients who had DM-related lower extremity amputations had regular rigorous foot examinations, while 29% had no examinations at all [10]. Therefore, early detection through prediction can provide information that will guide interventions to reduce the occurrence of foot ulcers and associated morbidity, mortality, and healthcare-related costs. The most widely used method for diagnosing neuropathy in a clinical setting involves testing the sensation of plantar foot tissue with a 10 g monofilament and tuning fork [11]. However, these tests may not be effective in identifying the early stages of neuropathy. Additionally, they are influenced by the subjectivity of the patient or clinician administering them [11,12,13]. More reliable techniques for diagnosing neuropathy, such as nerve conduction studies, are time-intensive and costly [13,14]. Recently, several diagnostic tests, including application of adhesive patches to measure skin hydration [15] and the use of infrared cameras to measure the foot’s thermal response to cold stimulus, have been proposed [16]. However, these techniques may also be both time-intensive and cost-prohibitive. As the number of patients with DM is still on the rise, it is imperative to predict the progression of DM in lower extremities.

A reliable, readily available, and cost-effective method to diagnose neuropathy can improve the accuracy and timeliness of detection [13]. Such a method may drive the development of effective interventions that can be applied to prevent ulcers and amputations. Early diagnosis of diabetic peripheral neuropathy will reduce the burden of diseases associated with it to the patient and the healthcare system. It was shown that measuring plantar pressures is an essential step in determining the response of tissues to stress during ambulation because it is considered to be a “surrogate measure of trauma, and an important contributing factor to skin breakdown” in patients with DM [17]. Several researchers have conducted clinical investigations to determine the variations in plantar pressures and stresses in patients with DM, since experimental data can provide insight on where high plantar pressures are applied and their potential effect on gait and development of foot ulcerations [18,19,20,21,22]. For example, the plantar tissue stress levels in the feet of patients with DM were compared for those with and without a history of foot ulcers. Results identified significantly decreased activity and higher stress levels in the DM patients with neuropathy compared to both DM patients without neuropathy and a control group [19]. It was suggested that DM patients with foot ulcers are at higher risk of plantar tissue injury even at relatively low levels of cumulative tissue stress [20]. Investigators have used wavelet transforms and neural-network classifiers with static plantar pressure measurements to classify individuals with diabetic neuropathy (DN), DM, or as non-diabetics [21,22]. Additionally, supervised machine learning algorithms and dynamic plantar pressure parameters have been used to successfully differentiate individuals with DN and DM from healthy individuals [23]. The researchers implemented binary support vector machine (SVM) classifiers to classify participants as individuals with diabetes with and without DN or as healthy controls. However, the literature shows that individuals with pre-diabetes (PD) have altered plantar pressure measures similar to those of individuals with DM when compared to healthy individuals, which can lead to individuals with PD being misclassified as having DM, or vice versa [24]. 

The purpose of this study was to determine the feasibility of using supervised machine learning algorithms to accurately classify persons who are diagnosed with pre-diabetes, diabetes, or diabetes with peripheral neuropathy. It was hypothesized that by combining several plantar pressure features along with patient-specific features, classification efficacy suitable for clinical use can be achieved.

## 2. Materials and Methods

### 2.1. Instrumentation 

Medilogic^®^ pressure-measuring insoles (Schönefeld, Germany) were used in this study. Six European insole sizes were used: 35–36, 37–38, 39–40, 41–42, 43–44, and 45–46. These insoles have a grid of sensors, ranging between 93 and 162 sensors depending on the size of the insole (Figure 1). The 0.75 × 1.5 cm rectangular sensors measure the change in electrical resistance, which is proportional to the applied normal pressure. The output of the sensors is a 0–255 digital scale, which is converted to 0–64 N/cm^2^ based on linear interpolation [25]. The sampling frequency of the insoles was set at 60 Hz.

### 2.2. Methods 

Data were collected at either the UNLV Diabetes and Endocrinology Clinic or at the UNLV Sports and Injury Research Center. The research protocol was approved by the Institutional Review Board at the University of Nevada, Las Vegas (IRB# 777036). Based on the size of related studies [21,23], a convenience sample of 110 individuals was recruited into three groups: Pre-DM (PD),DM without peripheral neuropathy (D), andDM with peripheral neuropathy (DN).

Participants were grouped into these three groups based on the diagnosis by their treating physicians. 

Participants were briefed about the testing protocol and asked to provide informed consent. Informed consent was obtained from all subjects involved in the study. Individuals who had pre-DM and DM with or without neuropathy were invited to participate. Exclusion criteria for the study were inability to walk without assistance or support, unhealed foot ulcerations, or pregnancy beyond second trimester. The demographic data of the participants are listed in Table 1.

Participants were fitted with the Medilogic^®^ insoles that best matched their foot size held by a pair of thin socks. The participants performed the following sequence of tasks for the walking trials: (1) sitting in a chair with both feet raised off the ground for 5 s; (2) standing up and staying still for 10 s; (3) sitting in the chair again with feet raised off the ground for 5 s; and (4) standing up and walking along a walkway (6–10 m) at their self-selected pace. The data were recorded when the participant reached a steady walking speed [26], as shown in Figure 2. Each participant was directed to perform five successful walking trials during the experiment. 

### 2.3. Preprocessing 

The sensor output data recorded during the walking trials were saved and processed with custom Matlab^®^ (Mathworks, Natick, MA, USA) algorithms. Support phase pressure data were extracted and processed to obtain plantar pressure–time histories for each step in each trial. The data were filtered using a 5 Hz low-pass filter [17,27]. To address the anatomical structure and function of the foot relative to the main phases of the support phase of walking, the foot was divided into three regions by dividing the insole into three sections:
Rearfoot (RF),Midfoot (MF), andForefoot (FF).

Sensors within each of these three regions of an insole size were grouped as shown in Figure 3, and the number of sensors in each region are shown in Table 2. 

For each foot (left/right) and each region *i*, the peak plantar pressure (PPP) was recorded at each time instant. PPP was defined as:(1)$${PPP}_{l,i}=max\left(\left(p_{j,t,k}\right)\right)$$
where *l* is limb (left (1)/right (2)), *i* is the region number, *t* is a time instant, and $p_{j,t,k}$ is pressure recorded by sensor *j* at time instant *t* of step *k*. Rearfoot, midfoot, and forefoot were labeled as 1, 2, and 3, respectively.

Similarly, the second set of extracted features was the peak pressure gradient (PPG) [28], which was computed at the location of the peak pressure sensor using the following equation: (2)$${PPG}_{l,i}=max\left(\frac{p_{j,t,k}-p_{n,t,k}}{\Delta_{n,t,k}}\right)$$
where, $p_{n,t,k}$ is pressure recorded at each of the eight sensors neighboring the peak pressure sensor $p_{j,t,k}$ of region *i* at time *t* of step *k* of the trial, and $\Delta_{n,t,k}$ is the length of the vector connecting the center of the peak pressure sensor to each of its eight neighbors. These distances are functions of the size of the insole sensors: 0.75 × 1.5 cm. The magnitude of PPG was determined to be the maximum spatial change at the location of the peak pressure sensor.

The third set of extracted features was the pressure–time integral (PTI) for each region, which was computed by summing the pressures of all of the sensors in each region over each time instant for each participant using the following equation:(3)$${PTI}_{l,i}=\sum_{t=0}^{T}\left(\sum_{j\in 1}^{s_{i}}p_{j,t,k}\right)$$where *t* is the time instant recorded within a step, *j* is sensor number, and $s_{i}$ is the number of active sensors in region *i*.

From these equations, features corresponding only to the left foot were used as features or,
(4)$$\begin{array}{ccc} \left\{F_{1,i}F_{2,i}F_{3,i}\right\}=\left\{{PPP}_{1,1:3}{PPG}_{1,1:3}{PTI}_{1,1:3}\right\} & & \;i=1,\;2,\;3 \end{array}$$

Additional features were introduced to assess the left/right asymmetry of the pressure features, which may be indicative of diabetic peripheral neuropathy. These features were defined as:(5)$$\left\{F_{4,i}F_{5,i}F_{6,i}\right\}=\left\{\frac{\left|{PPP}_{1,i}-{PPP}_{2,i}\right|}{{PPP}_{1,i}},\frac{\left|{PPG}_{1,i}-{PPG}_{2,i}\right|}{{PPG}_{1,i}},\frac{\left|{PTI}_{1,i}-{PTI}_{2,i}\right|}{{PTI}_{1,i}}\right\}\;i=1,\;2,\;3$$

Therefore, a total of 18 pressure features were extracted: the PPP, PPG, and PTI for each of the three regions for the left foot, along with the corresponding asymmetry indices computed using Equation 5. These features were combined with the four non-pressure features: age, mass, height, and HbA1c (Table 1) to train, validate, and test the models using various supervised machine learning algorithms. These features were labeled as:(6)$$\left\{F_{7,1}F_{7,2}F_{7,3}F_{7,4}\right\}=\left\{Age\;Mass\;Height\;HbA1c\right\}$$

Some participants completed fewer than five valid trials. For these participants, the total number of completed trials was used in the analysis. Additionally, applying the trial exclusion criteria discussed above resulted in a dataset consisting of 1621 observations or steps. 

### 2.4. Feature Selection and Creation of Feature Subsets

A total of 22 features were included in the research: 18 pressure-related and four non-pressure features. The 18 pressure features included the values of PPP, PPG, and PTI for the three regions of the left foot, as well as their corresponding asymmetry indices. The four non-pressure features included age, mass, height, and HbA1c. 

Multiple feature subsets were created using a combination of all or some of the 18 pressure and the four non-pressure features. The features that have the most impact on classifying the observations, and thereby are most useful for practical applications, can be determined by analyzing the performance of the models trained using these datasets of feature subsets.

Instead of studying the dataset with all features, it was decided to study various cases with various subsets to understand the importance of the different features in classification. Therefore, a total of 14 aggregated datasets were explored with various combinations of pressure and non-pressure features. In all of these datasets, PD, D, and DN were labeled as 0, 1, and 2 classes, respectively. The datasets were created by determining all possible combinations of the different pressure features, with the non-pressure features being included in each subset. HbA1c was of special interest, as some theorize it may not be a useful feature since some participants at an advanced stage of DM may have moderate HbA1c values if they have their blood sugar level under control. Consequently, HbA1c was included in the odd-numbered datasets only. Table 3 lists the features included in each dataset for training the models. 

### 2.5. Classification

Each of the 14 datasets were used to train the 31 algorithms that are available within the Classification Learner App in MATLAB version R2022a (Mathworks, Natick, MA, USA), (Appendix A). The first 20% of the data were set aside for testing. The remaining 80% of the data were trained with 10 cross-validation folds, split at the individual level such that all observations related to a single individual’s trial were kept together. Models were also trained and tested using only the pressure features in the 14 datasets, as well as two datasets, 15 and 16, that tested only the non-pressure features (one with and one without HbA1c)**.** The Classification Learner App allowed the analysis of 31 different machine learning algorithms. These included various ensemble algorithms that had been documented as being more effective than traditional machine learning models [29]. 

Precision, recall, the F1 score, and the false negative rate for DN were reported. Precision is the percentage of positively classified predictions that are true positive; recall is the percentage of correctly classified positive predictions out of the total true positive predictions; and the F1 score is the harmonic mean of these two metrics. A high F1 score indicates that the model is successful at predicting the positive cases, as well as a high certainty that the positive predictions are true. All performance metrics were computed based on formulas provided in the literature [30].

Since the primary purpose of testing the different datasets was to determine which features had a greater impact on classification, a principal component analysis (PCA) was carried out as a method of dimensionality reduction. The analysis was set to determine the principal components that explained at least 99% of the variability in the datasets. The analysis was performed on each dataset after the data were first standardized, and then a covariance matrix of the features was computed to calculate the eigenvectors and eigenvalues that were used to determine the principal components [31]. PCA is a reliable method of data reduction that can remove insignificant data, which can also prevent the model from overfitting the data. If PCA used with the feature subsets is able to produce high accuracy, it may provide a more generalizable and efficient model. 

## 3. Results

The models trained using all 22 features (Dataset 1) showed that several algorithms yielded similar performance results for the test data (Table 4). The best performing algorithm along with the corresponding precision, recall, F1 score, and percentage of false negatives for DN for each dataset are listed in Table 5. The best performing algorithm produced the highest test classification accuracy with the test data. The best performing algorithms for the models trained using only the pressure features in each dataset are listed in Table 6. When only considering the pressure features, the dataset pairs became identical. For clarity, only the odd datasets are listed. Table 7 lists the best performing algorithm, precision, recall, F1 score, and false negative rate for DN recorded for the datasets after PCA was applied. Both datasets 15 and 16 had precision, recall, and F1 scores of 100%.

## 4. Discussion

The purpose of this study was to determine the feasibility of using supervised machine learning algorithms to accurately classify persons who are diagnosed with pre-diabetes, diabetes, or diabetes with peripheral neuropathy. The classification accuracies, as well as the false negative rates for DN in the test dataset, were used to determine which algorithms and features subsets performed best. The false negative rates for DN participants are especially important because of the difficulty in diagnosing DN using current techniques. 

Table 5 indicates that all datasets yielded reasonably accurate classifications. However, since false negative rates are important for this type of analysis, it is reasonable to consider dataset 1 and datasets 11 through 14; all had zero percent false negative rates. Of these five datasets, datasets 13 and 14 had the highest F1 score, of 100%. The results may indicate that a subset of the tested features can be more successful in classification than using all available features. A recent study that used biomechanical data to train and test machine learning algorithms for DN diagnosis also found that using a subset of features, rather than the entire dataset, yielded greater accuracies [32]. Thus, the identification and inclusion of significant features, rather than all available features, could produce more effective and less computationally costly algorithms. 

The only pressure features in datasets 13 and 14 were for PTI. This indicates that PTI may play a significant role in DN classification. A previous study indicated that rearfoot values of peak pressure are significant in classification [22]. PPP has also been reported to be a significant indicator for DN [17,22]. This was corroborated in this study by the high performance of datasets 9 and 10, which relied solely on PPP pressure features, and datasets 5 and 6, which used PPP and PTI features. For these specific datasets, ensemble bagged trees and ensemble subspace KNN classifiers produced the best classifications of the data. It is also of note that subspace KNN was the second-best performing algorithm for five of the seven datasets where it was not the best performing, excepting only datasets 3 and 7. Table 5 also indicates that the exclusion of HbA1c did not yield any significant difference in results.

Testing only the non-pressure features, with and without HbA1c, resulted in 100% in all three measures. This may be because during data collection, while the pressure values would change for each trial, the non-pressure features for each participant would remain the same, since the measures were not variable during a single data collection session. As such, the current dataset did not produce any significant trends or results when looking only at the non-pressure features. No trends in the non-pressure features when plotted across the three classes were found, either. Since the non-pressure features did not have any discernable pattern in delineating the three classes, it could not be concluded that non-pressure features are all that is needed to classify individuals with PD, D, or DN, and further exploration into the impact of these features on classification was unfruitful. The non-pressure features alone were thus concluded to not have the ability to accurately classify the data. Analyzing solely the pressure features allowed the features that played a more significant role in accurate classification to become clearer.

The results of analyzing only the pressure features (Table 6) indicate that datasets 1, 5, 7, and 13 had test accuracies of greater than 80% and false negative rates of less than 3%. It should be noted that dataset 1 included all pressure features; dataset 5 had PPP and PTI features; dataset 7 combined PPG and PTI features; and dataset 13 only contained PTI features. The recurrence of PTI indicated its significance in classifying the progression of diabetes. When considering only pressure features, the results indicated that including more features does produce a higher test accuracy, but only when PTI features are included. For example, dataset 3, which included PPP and PPG features, achieved a test accuracy of 65% and false negative rate of 39%. Similarly, dataset 9 included only PPP features, and dataset 11 only included PPG features. All of these datasets produced test accuracies of less than 70% and false negative rates of greater than 50%. Overall, the inclusion of PTI, even as the sole pressure feature, greatly enhanced all three measures. The results from Table 6 indicate that a variety of pressure features used alone can still provide adequately accurate classifications, as datasets 1 and 5 had precision, recall, and F1 scores of higher than 90%. By comparing the results from Table 5 and Table 6, however, it can be seen that if using only a limited number of pressure features, having non-pressure features will greatly enhance the feasibility of this method, exemplified best by the drastic performance difference in dataset 11.

A study that used center of pressure data to train and test a deep clustering model found that individuals with DN tend to apply less force while walking and have longer stance times during gait [33]. As such, individuals who have DN might have values for PPG or PPP similar to those of the other classes, but their PTI values are more distinct, since the feature combines pressure, foot size (due to the increased number of sensors for larger insoles), and time. Additionally, since PTI represents the combined effect of these three factors, it can avoid the variability due to gait variation better than PPP or PPG. Given the high presence of PTI in the best performing algorithms, further study into the relationship between foot size, walking speed, and PTI is desirable.

Table 7 shows that PCA achieved comparable results for the best datasets of Table 5: datasets 1 and 11 through 14. The results that were achieved using ensemble classifier with subspace KNN model produced the highest results in conjunction with PCA. Apart from datasets 8, 9, and 12–14, PCA also led to higher false negative rates. While dimensionality reduction was explored to see if the model performance could be enhanced, the results indicate that PCA was unsuccessful in this.

While ensemble classifier with bagged trees was a high performing algorithm throughout the results, various forms of KNN also performed well for half of the datasets (Table 5). KNN performing well has been documented before in a previous study that also analyzed multiple algorithms in classifying DN using EMG and gait data [34].

## 5. Conclusions

This paper aimed to determine the feasibility of using machine learning algorithms to accurately classify individuals with pre-diabetes, diabetes without neuropathy, and diabetes with neuropathy. Using plantar pressure features (PPP, PPG, PTI), along with the non-pressure features (age, mass, height, and HbA1c), 14 datasets with different feature combinations were trained to determine the combination of features that provided the highest overall accuracy with the lowest false negative rates for diabetic peripheral neuropathy.

It was determined that using multiple pressure features yielded high classification accuracies. Though non-pressure features provided no visible trends that could be used to differentiate between the three classes, the addition of non-pressure features significantly contributed toward improving the model’s classification ability. The results show that PTI features played a stronger role in classification of the three subject groups, with the cases that included PTI yielding test accuracies of greater than 90%. On the other hand, the best case without PTI had a test accuracy of about 70%. Additionally, cases including PTI features had false negative rates of less than 3%. These observations are reasonable, since PTI combines pressure, foot size, and time. Since it represents the combined effect of these three factors over an area, it avoids the potential anomalies in peak pressure and pressure gradients that may be due to gait variations. It was also observed that the lowest false negative rate from a dataset that did not include PTI was 39.5%. The results also show that including HbA1c in the features consistently improved the results slightly.

The cubic SVM model had a test classification accuracy of 98.5% with no false negative classifications when using all of the available features. However, when classifying with a subset of features, ensemble classifiers with bagged trees or subspace KNN models performed the best. Various forms of KNN also performed well in all the analyses, making KNN an attractive algorithm for further analysis.

The effect of using PCA to reduce the dimensionality of the data was minimal. Overall, PCA was successful, though it led to higher false negative rates in nine of the 14 datasets.

This study was limited by the relatively small sample size, which led to an imbalanced dataset. A larger dataset with a more even split between classes would create a more generalizable model. Additionally, participants were asked to wear socks in this study to eliminate the uncertainty that could be caused by the type of shoe, but this does not represent daily life. A follow-up study that considers the effect of shoes on plantar pressure warrants investigation. Another limitation is that only HbA1c was included as a measure of DM severity. As mentioned before, HbA1c is a marker that can be controlled with a medication regime and lifestyle choices and is not always indicative of an individual’s diabetic progression. A future study including different means to assess DM severity, such as the length of diagnosis, would be beneficial. Similarly, arch type, foot size, and walking speed should be considered in future studies.

Overall, the research indicates that supervised machine learning algorithms are well-suited for classifying the progression of DM. It would be of interest to repeat the study using a larger number of participants. 

## Figures and Tables

**Figure 1 sensors-23-04658-f001:**
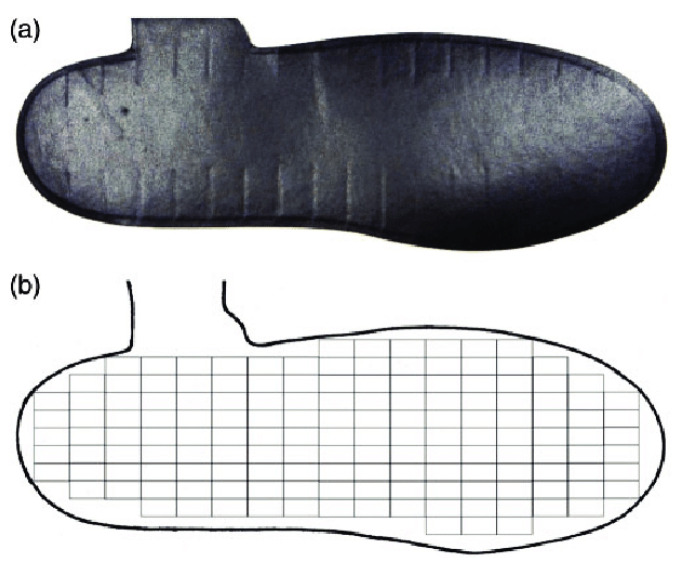
(**a**) A typical Medilogic insole. (**b**) Corresponding sensor map.

**Figure 2 sensors-23-04658-f002:**
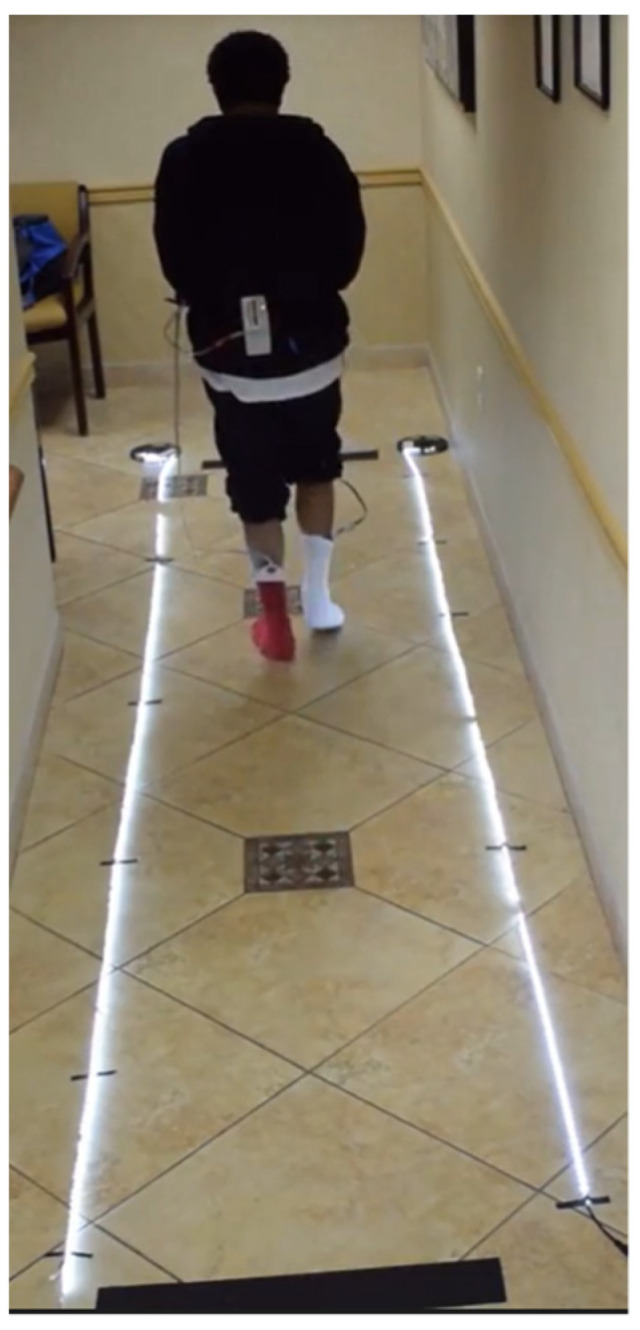
A participant fitted with socks conducting trial.

**Figure 3 sensors-23-04658-f003:**
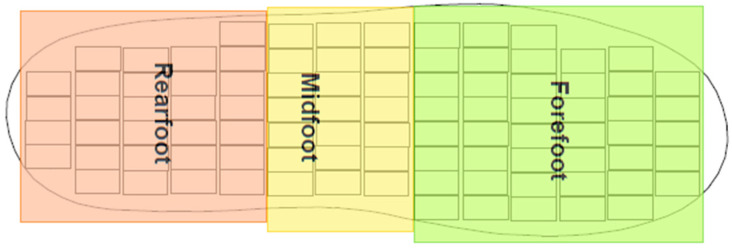
Representation of the typical division of the plantar foot overlaid on a size 35 insole map.

**Table 1 sensors-23-04658-t001:** Participant demographic data for the pre-DM (PD), DM without peripheral neuropathy (D), and DM with peripheral neuropathy (DN) groups (mean ± standard deviation).

	PD	D	DN
No. of participants	19	62	29
Sex (M/F)	9/10	29/33	14/15
Age (Years)	59.6 ± 11.4	58 ± 15.8	63.8 ± 10.2
Body Mass (kg)	93.7 ± 27.6	90.8 ± 26.2	96.9 ± 25.9
Height (m)	1.68 ± 0.09	1.66 ± 0.09	1.70 ± 0.10
HbA1c (%)	5.9 ± 0.4	7.6 ± 1.6	7.2 ± 1.3

**Table 2 sensors-23-04658-t002:** Number of sensors in each foot region, for each insole size.

European Insole Size	Region	Number of Sensors
35–36	RF	29
	MF	29
	FF	35
37–38	RF	38
	MF	32
	FF	37
39–40	RF	34
	MF	32
	FF	50
41–42	RF	45
	MF	34
	FF	50
43–44	RF	47
	MF	38
	FF	66
45–46	RF	48
	MF	36
	FF	78

**Table 3 sensors-23-04658-t003:** List of datasets with the features ($F_{i,j}$) included: Pressure features (*i* = 1:6) at the regions (*j* = 1:3): rearfoot (1), midfoot (2), and forefoot (3) regions, and non-pressure features (*i* = 7; *j* = 1:4).

Training Dataset	Description	No. of Features	Features
1	Pressure Features: PPP, PPG, PTI, Corresponding Asymmetry	18	$F_{1,1},F_{1,2},F_{1,3},F_{2,1},F_{2,2},F_{2,3},F_{3,1},F_{3,2},F_{3,3}$ $F_{4,1},F_{4,2},F_{4,3},$ $F_{5,1},F_{5,2},F_{5,3},F_{6,1},F_{6,2},F_{6,3}$
	Non-Pressure Features	4	$F_{7,1},F_{7,2},F_{7,3},F_{7,4}$
2	Pressure Features: PPP, PPG, PTI, Corresponding Asymmetry	18	$F_{1,1},F_{1,2},F_{1,3},F_{2,1},F_{2,2},F_{2,3},F_{3,1},F_{3,2},F_{3,3}$ $F_{4,1},F_{4,2},F_{4,3},F_{5,1},F_{5,2},F_{5,3},$ $F_{6,1},F_{6,2},F_{6,3}$
	Non-Pressure Features	3	$F_{7,1},F_{7,2},F_{7,3}$
3	Pressure Features: PPP, PPG, Corresponding Asymmetry	12	$F_{1,1},F_{1,2},F_{1,3},F_{2,1},F_{2,2},F_{2,3},F_{4,1},F_{4,2},F_{4,3}$ $F_{5,1},F_{5,2},F_{5,3}$
	Non-Pressure Features	4	$F_{7,1},F_{7,2},F_{7,3},F_{7,4}$
4	Pressure Features: PPP, PPG, Corresponding Asymmetry	12	$F_{1,1},F_{1,2},F_{1,3}$ $,F_{2,1},F_{2,2},F_{2,3}$ $,\;F_{4,1},F_{4,2},F_{4,3}$ $F_{5,1},F_{5,2},F_{5,3}$
	Non-Pressure Features	3	$F_{7,1},F_{7,2},F_{7,3}$
5	Pressure Features: PPP, PTI, Corresponding Asymmetry	12	$F_{1,1},F_{1,2},F_{1,3},F_{3,1},F_{3,2},F_{3,3},F_{4,1},F_{4,2},F_{4,3}$ $F_{6,1},F_{6,2},F_{6,3}$
	Non-Pressure Features	4	$F_{7,1},F_{7,2},F_{7,3},F_{7,4}$
6	Pressure Features: PPP, PTI, Corresponding Asymmetry	12	$F_{1,1},F_{1,2},F_{1,3},F_{3,1},F_{3,2},F_{3,3},F_{4,1},F_{4,2},F_{4,3}$ $F_{6,1},F_{6,2},F_{6,3}$
	Non-Pressure Features	3	$F_{7,1},F_{7,2},F_{7,3}$
7	Pressure Features: PPG, PTI, Corresponding Asymmetry	12	$F_{2,1},F_{2,2},F_{2,3},F_{3,1},F_{3,2},F_{3,3},F_{5,1},F_{5,2},F_{5,3}$ $F_{6,1},F_{6,2},F_{6,3}$
	Non-Pressure Features	4	$F_{7,1},F_{7,2},F_{7,3},F_{7,4}$
8	Pressure Features: PPG, PTI, Corresponding Asymmetry	12	$F_{2,1},F_{2,2},F_{2,3},F_{3,1},F_{3,2},F_{3,3},F_{5,1},F_{5,2},F_{5,3}$ $F_{6,1},F_{6,2},F_{6,3}$
	Non-Pressure Features	3	$F_{7,1},F_{7,2},F_{7,3}$
9	Pressure Features: PPP and Corresponding Asymmetry	6	$F_{1,1},F_{1,2},F_{1,3},F_{4,1},F_{4,2},F_{4,3}$
	Non-Pressure Features	4	$F_{7,1},F_{7,2},F_{7,3},F_{7,4}$
10	Pressure Features: PPP and Corresponding Asymmetry	6	$F_{1,1},F_{1,2},F_{1,3},F_{4,1},F_{4,2},F_{4,3}$
	Non-Pressure Features	3	$F_{7,1},F_{7,2},F_{7,3}$
11	Pressure Features: PPG and Corresponding Asymmetry	6	$F_{2,1},F_{2,2},F_{2,3}$ $F_{5,1},F_{5,2},F_{5,3}$
	Non-Pressure Features	4	$F_{7,1},F_{7,2},F_{7,3},F_{7,4}$
12	Pressure Features: PPG and Corresponding Asymmetry	6	$F_{2,1},F_{2,2},F_{2,3},F_{5,2},F_{5,3}$
	Non-Pressure Features	3	$F_{7,1},F_{7,2},F_{7,3}$
13	Pressure Features: PTI and Corresponding Asymmetry	6	$F_{3,1},F_{3,2},F_{3,3},F_{6,1},F_{6,2},F_{6,3}$
	Non-Pressure Features	4	$F_{7,1},F_{7,2},F_{7,3},F_{7,4}$
14	Pressure Features: PTI and Corresponding Asymmetry	6	$F_{3,1},F_{3,2},F_{3,3},F_{6,1},F_{6,2},F_{6,3}$
	Non-Pressure Features	3	$F_{7,1},F_{7,2},F_{7,3}$
15	Non-Pressure Features	4	$F_{7,1},F_{7,2},F_{7,3},F_{7,4}$
16	Non-Pressure Features	3	$F_{7,1},F_{7,2},F_{7,3}$

**Table 4 sensors-23-04658-t004:** Precision, recall, F1 score, and false negative rates for the best performing algorithms, ranked by test accuracy for Dataset 1.

Algorithm	Precision (%)	Recall (%)	F1 Score	False Negative Rate (%)
Cubic Support Vector Machine (SVM)	97.9	98.5	98.2	0
Subspace K-Nearest Neighbors (KNN) *	96.9	98.5	97.7	1.9
Bagged Trees *	98.0	96.8	97.4	1.9
Wide Neural Network	96.4	98.4	97.4	1.9
Boosted Trees	97.6	96.4	97.0	1.9
Fine KNN	96.6	96.7	96.6	2.8
Weighted KNN	96.0	97.2	96.6	0.9
Fine Tree	96.3	95.4	95.9	1.9
Medium Neural Network	94.2	97.3	95.7	0
Trilayered Neural Network	94.9	95.6	95.2	2.8
Quadratic SVM	94.0	95.4	94.7	0
RUSBoosted Trees *	96.6	93.1	94.8	1.9

* Ensemble.

**Table 5 sensors-23-04658-t005:** Best performing algorithm for each dataset with precision, recall, F1 score, and false negative rate.

Dataset	Best Performing Algorithm	Precision (%)	Recall (%)	F1 Score	False Negative Rate (%)
**1**	Cubic SVM	97.9	98.5	98.2	0
**2**	Subspace KNN *	98.3	98.7	98.4	2.4
**3**	Bagged Trees *	98.4	99.3	98.7	4.7
**4**	Bagged Trees *	98.2	99.5	98.8	0
**5**	Subspace KNN *	99.0	99.2	99.1	2.3
**6**	Subspace KNN *	99.2	99.2	99.2	2.3
**7**	Bagged Trees *	98.5	98.5	98.5	3.5
**8**	Subspace KNN *	98.0	98.9	98.5	3.5
**9**	Bagged Trees *	100	100	100	0
**10**	Subspace KNN *	98.6	99.5	99.0	2.4
**11**	Bagged Trees *	99.2	99.4	99.3	0
**12**	Bagged Trees *	99.8	99.6	99.7	0
**13**	Subspace KNN *	100	100	100	0
**14**	Subspace KNN *	100	100	100	0

* Ensemble.

**Table 6 sensors-23-04658-t006:** Best performing algorithm, precision, recall, F1 score, and false negative rate for each dataset with only pressure features.

Dataset	Best Performing Algorithm	Precision (%)	Recall (%)	F1 Score (%)	False Negative Rate (%)
**1**	Fine KNN	91.2	92.3	91.7	0
**3**	Fine KNN	64.6	65.1	64.8	39.5
**5**	Quadratic SVM	92.7	93.4	93.1	0
**7**	Medium Neural Network	88.6	89.2	88.9	2.4
**9**	Weighted KNN *	60.9	64.1	62.5	52.3
**11**	Bagged Trees *	49.9	57.5	53.4	67.1
**13**	Fine Tree	82.9	88.6	85.7	2.4

* Ensemble.

**Table 7 sensors-23-04658-t007:** Best performing algorithm, number of components, precision, recall, F1 score, and false negative rate for each dataset with features selected through PCA.

Dataset	Algorithm	Number of Components	Precision (%)	Recall (%)	F1 Score (%)	False Negative Rate (%)
**1**	Subspace KNN *	9	93.1	95.5	94.2	4.7
**2**	Subspace KNN *	9	94.4	94.7	94.6	4.7
**3**	Subspace KNN *	8	93.0	93.8	93.4	5.8
**4**	Subspace KNN *	8	95.5	96.7	96.1	3.5
**5**	Subspace KNN *	6	96.5	97.1	96.8	8.1
**6**	Subspace KNN *	6	96.9	97.5	97.2	4.7
**7**	Subspace KNN *	7	94.8	95.6	95.2	5.9
**8**	Bilayered Neural Network	6	97.3	96.2	96.8	1.2
**9**	Subspace KNN *	6	97.9	98.0	98.0	1.2
**10**	Bagged Trees *	6	95.5	97.4	96.4	3.5
**11**	Bilayered Neural Network	6	96.1	97.1	96.6	8.2
**12**	Subspace KNN *	6	94.8	95.5	95.2	1.2
**13**	Subspace KNN *	5	100	100	100	0
**14**	Subspace KNN *	5	100	100	100	0

* Ensemble.

## Data Availability

The data presented in this study will be available in the future upon request to the authors.

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
