# Peer review of "Prediction of Diabetes Mellitus Progression Using Supervised Machine Learning"

_sensors, 2023, doi:10.3390/s23104658_

Round 1
Reviewer 1 Report
This work performed a supervised machine learning with the three groups of participants to predict diabetes mellitus. Matlab is a major tool for the classification. The results of prediction were achieved with high accuracy. However, there are several concerns and they can be improved.
1. Please clearly mention the inclusion criteria of this study.
2. How are the numbers of participants obtained? Was the sample size calculation performed?
3. Why did this study ask the participant to wear socks to hold the insoles? Why did this study not ask the participants to wear shoes which is similar to a daily life?
4. Why did this study raise 14 datasets? Please explain the concept of picking up these datasets.
5. There are several performance metrics such as precision, sensitivity, F1 score and AUC. Did the authors perform these performance metrics? These metric are interesting and important to show.
6. Does the severity of DM affect the prediction? Also, the severity of DM can influence or alter the results presented by this study.
7. The authors should discuss the regions of foot to the results that obtained. Because this study divided foot into 3 regions which might be interesting if this study can further investigate.
8. Please add more discussion which compared to other current studies, during 5 years, that predicted diabetic neuropathy.
9. Are there other limitations of this study, excluding a larger number of participants?
10. Line 37, please update the recent statistics of US people who have DM. The statistics of 2019 are reported on https://diabetes.org Furthermore, the citation that is cited seems not correct. It should be Reference no. 4, not no. 3.
11. Line 39, please check the citation.
12. Line 160, it is better to delete this line. Table 2 header was already shown on Line 168.
13. Line 186, what does "active?" mean?
14. Line 305, the specific table number is missing. It should be Table 7.
15. Line 324, it should be "On the other hand" not "One the other hand".
Reviewer 2 Report
(1) The research title is concise and clear for a study. Is this is a result based paper? If yes the the title should be considered agin.
(2) Clarify about 18 pressure features (subsection 2.4)
(3) Justify selection of PCA for dimensionality reduction
(4) When earlier accuracies are good achieving good accuracy then may you suggest best among them
(5) Many papers we may find in the literature about Diabetics. Some papers may be added in this paper:
Khan, Y. F., Kaushik, B., Chowdhary, C. L., & Srivastava, G. (2022). Ensemble Model for Diagnostic Classification of Alzheimer’s Disease Based on Brain Anatomical Magnetic Resonance Imaging. Diagnostics, 12(12), 3193.
Reviewer 3 Report
In this work, the authors proposed constructing pressure features with dynamic plantar pressure measurements and applying machine learning models for diabetic peripheral neuropathy prediction. Various feature combination, ML models were compared and evaluated based on the metrics of accuracy and false negative rate.
Major weakness:
Scientific soundness
1. When the authors performed train-test split, please clarify whether the split is at individual level or at observation level? Would observations from the same individual be split into both training and testing data? If so, there would be information leak, and the performance would be over-estimated.
2. Table 4-7, since the data set is imbalanced, accuracy is not a good measurement for model performance, I suggest adding precision, recall and F1 score as evaluation metrics.
3. The authors should discuss small sample size in this study as a major limitation, instead of briefly mention it in Line 337. Small sample size could lead to large variation of the model performance on testing data. (My understanding is the testing data only have 100-200 observations? having 1 or 2 sample misclassified will cause a large change of model performance)
4. Formula (3), based on the formula, larger shoe size means more sensors in each region and higher values of PTI (because it’s a summation of the pressure at individual sensors). In addition, for people walking slower (longer time), the value of PTI will also be higher. Have the authors considered scaling this measurement to take account of these factors?
5. It also surprises me that the model performance with only PTI related pressure features (dataset 13 and 14) is the best. Is there any explanation for that? My take is that, this feature should not be regarded as a pressure feature only, it also measures walking speed and foot size, which may indicate that walking speed and foot size are important indicators for DN? If this is the case, what is more predictive, Pressure signals, shoes size, or walking speed? Could the authors quantify the effects/importance of individual factors? I suggest performing additional analysis and further discussing this feature.
6. Formula (5) by using absolute value, there will be information loss with respect to the directions. What’s the rational to use the absolute value of differences in the formula?
7. Line 233-236. What’s PCA used for in this study? Is it for dimension reduction. Here, there are only limited number of features (18+4), why is there a need for dimension reduction?
8. Based on Table 6, it seems that the non-pressure features are predictive (removing them make the model performance much worse). I suggest the authors build another model with non-pressure features only, in order to prove that the pressure related features are useful.
Other suggestions and questions:
9. Line 104, typo, should it be 43-44 instead of 43-55?
10. Page 4-5, there are 2 titles for Table 2. One of them should be deleted.
11. Line 173, what is step k, is the step correspond to the four tasks described in Line 135-138 or the number of trials described in Line 139-140?
12. Since p_n,t,k is measured at eight neighboring sensors, in formula (2), should this be an average/other form of aggregation of the eight measurements?
13. Formula (3) is a little confusing, do you mean t from 0 to T. Below the summation sign should be t=0, and above the sign should be T.
14. Line 186, typo, there is a “?”.
15. Line 228, since 20% of the samples are set aside for testing purpose, what’s the purpose of performing a 10-fold cross-validation, is cross-validation used for hyperparameter tuning?
16. Line 230-231. Since you have 3 groups (PD, D and DN), why not multiclass classification, instead of binary classification?
17. Line 305, typo, it should be “Table 7”, instead of “Table ”.
Minor editing of English language required.
Round 2
Reviewer 1 Report
Authors have revised following the comments. However, there are some minor points need to be corrected or checked.
1. There are too much space before Table 2.
2. There is a black square at the equation (3).
3. It is better to put the Table 3's title to be on the same page with table.
4. There are space at Line 345-346.
5. Which table does the authors mean on the Line 347, the table's number is missing?
Reviewer 3 Report
I thank the authors for making efforts to address my comments. The revision improves with respect to many issues I have raised in my review. However, I have some concerns, particularly with respect to point 4, 5 and 8:
Point 8: Based on Table 6, it seems that the non-pressure features are predictive (removing them make the model performance much worse). I suggest the authors build another model with non-pressure features only, in order to prove that the pressure related features are useful.
Author: Thank you for the feedback. Due to the nature of the data, while the pressure features varied slightly for each observation, the non-pressure features were largely the same for all corresponding observations. Due to this, testing a model using only non-pressure data results in 100% accuracy, but that is due to the invariable nature of the data. By testing the pressure features, we were able to conclude that using a variety of pressure features together yields high results, as can be seen by the highest performing dataset in Table 6 being dataset 1.
Comment: Could you elaborate what do you mean by “the non-pressure features were largely the same for all corresponding observations”? Do you mean, for example, the DN subjects are generally older or has higher mass than PD and D subjects in your sample (as seen in Table 1), which makes it possible to create a perfect separation of the classes with these non-pressure features? Why is this the case? Is it sampling bias? or is it representative of the population, i.e., generalizable? If it's sampling bias, I doubt about the conclusion “the addition of non-pressure features significantly contributed toward improving the model’s classification ability.” If the pattern of perfect separation (these 4 non-pressure features predict DN perfectly) is generalizable, why is there a need to use pressure features? I suggest the authors further discuss this. Finally, when these non-pressure features are highly predictive in your sample, I’m not sure how meaningful the comparisons are in your result Table 5, as the high scores will be mostly driven by these non-pressure features and the differences may just be due to randomness.
Point 4: Formula (3), based on the formula, larger shoe size means more sensors in each region and higher values of PTI (because it’s a summation of the pressure at individual sensors). In addition, for people walking slower (longer time), the value of PTI will also be higher. Have the authors considered scaling this measurement to take account of these factors?
Author: Thank you for the comment. While the observations of the Reviewer are correct, we included body mass as a feature. We think this may lead to a more meaningful analysis compared to normalizing pressure by dividing it by mass.
Point 5: It also surprises me that the model performance with only PTI related pressure features (dataset 13 and 14) is the best. Is there any explanation for that? My take is that, this feature should not be regarded as a pressure feature only, it also measures walking speed and foot size, which may indicate that walking speed and foot size are important indicators for DN? If this is the case, what is more predictive, Pressure signals, shoes size, or walking speed? Could the authors quantify the effects/importance of individual factors? I suggest performing additional analysis and further discussing this feature.
Author: As the Reviewer observed, PTI combines pressure, foot size, and time. Since it represents the combined effect of these three factors over an area, it avoids the potential anomalies in peak pressure and pressure gradients that may be due to gait variations. We agree that looking closely at foot size and walking speed should be considered in future studies. Arch type might also be a measurement of interest.
Comment: Point 4 and 5 are similar issues with the one I raised as Point 8. PTI combines pressure, foot size, and time. It could be a proxy of the non-pressure feature of Mass and Height. For the conclusion of “PTI play a significant role in DN classification”, could it be simply due to Mass and Height features are highly predictive in your sample? Is this generalizable? How could you justify the usefulness of the pressure features?
Point 15: Line 228, since 20% of the samples are set aside for testing purpose, what’s the purpose of performing a 10-fold cross-validation, is cross-validation used for hyper-parameter tuning?
Author: Thank you for the comment. We included the 10-fold cross-validation along with the holdout method so that the model was repeatedly trained with varying datasets, rather than simply the one time with 80% of the data. In this way, the model could be more generalizable and less reliant on a single data set.
Comment: Do you mean repeated cross-validation with 4:1 split for training and testing? 10-fold cross-validation usually refers to 9:1 split.
Point 17: Line 305, typo, it should be “Table 7”, instead of “Table ”.
Author: Thank you!
In the revised version, I still see “Table ” in line 347.
Additional suggestions:
1. Line 242-243, the definition of recall is incorrect. Recall is the fraction of actual positive that are predicted to be positive.
Minor editing of English language required.
